# Neuroticism Overestimated? Neuroticism Versus Hypertonia, Pain and Rehabilitation Outcomes in Post-Spinal Cord Injury Patients Rehabilitated Conventionally and with Robotic-Assisted Gait Training

**DOI:** 10.3390/brainsci14111153

**Published:** 2024-11-18

**Authors:** Alicja Widuch-Spodyniuk, Beata Tarnacka, Bogumił Korczyński, Aleksandra Borkowska

**Affiliations:** 1Research Institute for Innovative Methods of Rehabilitation of Patients with Spinal Cord Injury in Kamien Pomorski, Health Resort Kamien Pomorski, 72-400 Kamien Pomorski, Poland; alicjamariawiduch@gmail.com (A.W.-S.); b.korczynski@u-kp.pl (B.K.); 2Department of Rehabilitation, Eleonora Reicher National Institute of Geriatrics, Rheumatology and Rehabilitation, 02-637 Warsaw, Poland; 3Department of Rehabilitation, Medical University of Warsaw, 02-091 Warsaw, Poland; aleksandra.borkowska@wum.edu.pl

**Keywords:** spinal cord injury (SCI), neuroticism, spasticity, pain

## Abstract

Background: The aim of the present study was to analyse the association between neuroticism (one of the Big Five personality traits) and the most common secondary sensorimotor complications occurring in patients after spinal cord injury (SCI), i.e., muscle spasticity (hypertonia) and pain, and to investigate the associations between neuroticism and the effects of conventional rehabilitation (dynamic parapodium) and those using robotic-assisted gait training (RAGT) in this group of patients. In addition, the association of neuroticism with self-efficacy, personal beliefs about pain control, and adopted coping strategies among SCI patients was analysed. These data can be used as a reference for designing effective forms of therapy and support dedicated to this group of patients. Methods and procedures: Quantitative analysis included 110 patients after SCI. The participants were divided by simple randomisation into a rehabilitation group with RAGT and a rehabilitation group with dynamic parapodium therapy (DPT). The following survey instruments were used for data collection: Revised NEO Personality Inventory (NEO-PI-R); Ashworth Scale; the Spinal Cord Independence Measure III (SCIM III); the Walking Index for Spinal Cord Injury II (WISCI-II); the American Spinal Injury Association Impairment Scale (AIS); the Pain Coping Strategies Questionnaire—CSQ; and the Beliefs about Pain Control Questionnaire—BPCQ. Outcomes and results: analyses showed a positive association between neuroticism and spastic tension (rho = 0.39; *p* < 0.001). Conclusions and implications: the study showed that a high level of neuroticism correlates with a higher level of spasticity, but no such correlation was observed for pain. Additionally, the study did not show a significant correlation between neuroticism and rehabilitation outcome depending on the rehabilitation modality (RAGT vs. DPT). The results underline the importance of carrying out a psychological diagnosis of patients to provide therapeutic support in the rehabilitation process.

## 1. Introduction

Spinal cord injury (SCI) is a traumatic event that has motor and sensory consequences [1]. The rehabilitation prognosis after SCI varies based on the severity, degree, and level of injury and its completeness [2,3].

The aim of rehabilitation is to assist the patient in returning to their optimal relative ability to function in key aspects of life. It is advisable that rehabilitation begins as soon as the patient’s condition stabilises after the injury. Due to the multitude of complications and consequences, rehabilitation should be carried out by a multispecialist patient-centred team based on the results of a functional diagnosis of functional impairment [1,3,4].

Due to the complexity of the problems experienced by SCI patients, researchers are constantly looking for effective therapeutic methods. In recent years, a number of methods have been developed to support rehabilitation, including robotic-assisted gait training (RAGT). RAGT is aimed at stimulating neurophysiological mechanisms. Compared to traditional methods, RAGT allows for longer and more intensive training with physiologically normal gait patterns [5]. The effectiveness of RAGT has yet to be fully confirmed due to the insufficient number of randomised clinical trials. A number of studies carried out on people with SCI or stroke show that the use of robots in rehabilitation leads to better or equal results in improving gait function compared to conventional methods such as dynamic parapodium training (DPT) [5,6,7,8]. In the realities of Polish health care services, dynamic parapodium training (DPT) is conventionally used [6]. Figure 1 illustrates how RAGT and DPT differ and apply [9].

Robots that are used in the rehabilitation of people with SCI include the Lokomat and the exoskeletons. The Lokomat is equipped with a treadmill, rigid frames attached externally to the patient’s lower limbs, and a harness [9,10]. The Lokomat allows for the adjustment of range of movement, speed, and amount of body weight loading, and has a positive effect on patients’ range of movement and mobility [11,12].

The exoskeleton is a mobile device equipped with lower limb orthoses to support movement on a flat surface, which allows gait training to be carried out with optimal intensity and repetition [13,14].

The effectiveness of rehabilitation depends on the knowledge and experience of the therapeutic team, the implementation of appropriate techniques, the use of targeted assistive equipment, as well as the characteristics of the patient themselves, including their somatic state; personality traits; cognitive, emotional, and behavioural aspects; and other coexisting factors [15,16]. It is worth taking into account the patient’s reaction to the traumatic event and the losses suffered, including, but not limited to, the degree of disability, uncertainty, the extent of adaptation to the demands of rehabilitation, the ability to cope with physical discomfort, the relationship with the medical staff, and other considerations.

Psychological factors are important in the context of RAGT use and human–robot interaction itself. Although these devices are safe, well adapted and accepted [17], some studies indicate that patients feel insecure and apprehensive about the exoskeleton [18]. However, other studies show that rehabilitation robots give patients more control, and a sense of impact and security [19].

Personality is a relatively fixed and stable structure of an individual’s traits and dispositions expressed in the way a person thinks, feels, and reacts. Personality allows an individual to adapt to their environment and also sets them apart from others [20]. The Five-Factor Personality Theory (PMO) of Costa and Mc Crae is one of the most widely studied personality models [21,22]. The PMO and the personality traits it distinguishes, i.e., neuroticism, extraversion, openness to experience, agreeableness, and conscientiousness, are related to aspects of health [23,24,25,26,27,28,29].

Neuroticism is linked to stress theory and has the greatest association with aspects of somatic health [30,31]. Neuroticism is a measure of the quality of adaptation and emotional balance. Individuals with high levels of this trait tend to sustain negative emotions (including fear, sadness, shame, anger, guilt, threat, and disgust), worry and ruminations, are highly excitable, have difficulties with impulse control and coping with stress in a constructive way, and often have low self-esteem.

Neuroticism and its components (sensitivity, cynicism, pessimism, anxiety, and depression) have been shown to be associated with an increased risk of death [31,32]. High neuroticism and low conscientiousness are associated with metabolic syndrome and risk factors for type II diabetes, obesity and cardiovascular disease [33,34]. These two traits have also been associated with higher levels of the pro-inflammatory cytokine interleukin-6 (IL-6) and C-reactive protein, a marker of chronic inflammation [29,35].

Neuroticism can increase pain sensitivity and intensify it, reduce patients’ resilience to pain, interfere with the ability to cope and increase the tendency to somatise [36,37,38,39,40]. A correlation has been shown between high neuroticism and musculoskeletal complaints, most notably back pain [41,42,43] and tension headaches [44].

Pain after spinal injury, as well as spasticity, are very common complications after SCI that may affect rehabilitation prognosis [45,46]. There is a lack of research on the potential relationship between neuroticism and muscle hypertonia (spasticity). Muscle spasticity, like pain, is a common and serious sensorimotor complication after SCI, causing increased tissue tension and stiffness, limiting or preventing movement [47,48].

Neuroticism is associated with a stress response and difficulty undertaking coping activities [49,50]. Psychological stress is listed as one of the so-called ‘harmful triggers’ of spasticity [51]. Neuroticism promotes a passive attitude towards health problems and difficulties in complying with rehabilitation recommendations [16]. It can negatively affect the rehabilitation process on many levels: from non-compliance and lack of motivation to pursue rehabilitation to the association with somatic complaints (pain, muscle tension, etc.) or their intensification. A study by Yue Xu et al. found a negative correlation between neuroticism and measures of knee rehabilitation results [16].

This study aimed to answer the following questions: (1) Is neuroticism related to the severity of pain in people at an early stage after SCI? (2) Is neuroticism related to spasticity severity in early post-SCI patients? (3) Is neuroticism related to the results of conventional rehabilitation and RAGT-guided rehabilitation in post-SCI patients? (4) Are neuroticism and its components related to self-efficacy, pain control beliefs and pain management strategies in post-SCI patients?

## 2. Materials and Methods

### 2.1. Setting and Sampling Method

This study was a part of a prospective clinical trial and was conducted in accordance with the principles of the Declaration of Helsinki. The project was approved by the Ethical Council of the Regional Chamber of Physicians in Szczecin (Poland) (No. OIL-Sz/MF/KB/452/05/07/2018; No. OIL-SZ/MF/KB/450/UKP/10/2018).

Patients were recruited through self-selection from the general population of people with SCI in Poland. (Patient recruitment lasted from 19 April 2018 to 13 December 2021).

All participants (*n* = 158) signed an informed consent form to participate in the study. Each underwent a diagnostic evaluation by a neurologist, medical rehabilitation physician, physiotherapists, and a psychologist before qualifying for the study.

Medical criteria for inclusion in the study included a period of 3 months to 2 years having passed since the injury was sustained; the patient had to be described as conscious, able to cooperate with the physiotherapist, and adapted to an upright body position; patients with both complete and incomplete SCI (in C, Th and L segments) with surgical stabilisation of the spine who were in the phase of completed bone fusion were eligible; patients were required to have no contraindications to rehabilitation, including venous thrombosis, pulmonary embolism, orthostatic hypotension, epilepsy or infections. Exclusion criteria included high complete tetraplegia, as well as very low lumbar spine injury; lack of complete bone fusion after spinal surgery and assumption of stabilisation; respiratory failure, circulatory failure grade III and IV according to the New York Heart Association (NYHA) criteria; osteoporosis; lower limb shortening > 2 cm; the presence of bedsores and deep abrasions or skin lesions; severe spasticity (4 points on the Ashworth Scale) or muscle contractures making robotic rehabilitation impossible; past or present neurological disorders, i.e., traumatic spinal stroke, multiple sclerosis, or infantile cerebral palsy; symptoms of recurrent autonomic dysreflexia; reduced level of intellectual functioning preventing completion of questionnaires and interviewing; age < 18 years. A total of 110 patients were included in the study; 48 patients did not pass the pre-qualification. Patients enrolled in the study were allocated by a blinded investigator to one of two study groups using randomisation by coin toss. Those who ended up in the S0 (control) group received conventional rehabilitation with 30 min sessions using DPT; in the S1 group (the experimental group) the RAGT (exoskeleton EKSO or Lokomat) was also used for 30 min sessions. The rehabilitation programme lasted seven weeks, with a one-week break taking place after three weeks.

### 2.2. Participants

A total of 110 patients, including 89 men and 21 women with neurological impairment after SCI, participated in the study. Patient groups were divided according to rehabilitation modality (S0 *n* = 31; S1 *n* = 79), range and completeness of injury assessed according to the International Standards for Neurological SCI Classification using the American Spinal Injury Society Impairment Scale (AIS): AIS A (*n* = 44) and AIS B,C,D (*n* = 66) and level of injury: cervical injury at cervical vertebrae C1-7 (*n* = 24) or thoracic vertebrae Th1-12 (*n* = 57) or sacral injury at lumbar vertebrae L-1-L5 (*n* = 29).

From those enrolled in the study, groups of people with high and low neuroticism were distinguished. The division was made on the basis of the results of the NEO-PI-R Inventory: people with low neuroticism obtained scores from 1 to 4 sten on the neuroticism scale, people with high neuroticism obtained scores from 7 to 10 sten (Table 1). People with medium levels of neuroticism were excluded from the analyses (*n* = 32; 29.1%).

The results of 78 individuals were analysed, the majority of whom were those with low levels of neuroticism. Table 2 shows the results on the characteristics of the study group.

### 2.3. Outcome Measures

The analysed variables were operationalised using research tools of proven relevance and reliability translated into Polish.

To measure the severity of neuroticism, the Revised NEO Personality Inventory (NEO-PI-R) was used [52]. The neuroticism scale consists of 5 subscales: anxiety (tendency to be anxious, afraid, worried, feeling tense; aggressive hostility (tendency to experience anger, malice, and frustration); depressiveness (experiencing feelings of guilt, anhedonia, alienation, and hopelessness); over-criticism (feeling worse than others, the tendency to self-evaluate, and the conviction of being judged badly by others); impulsivity (limited self-control over impulses) and hypersensitivity (the conviction of not being able to overcome stressful situations, dependence, and disorganisation in difficult situations) [52].

A modified version of the 5-point Ashworth Scale was used to measure muscle tone and spasticity, with 0 indicating no increase in muscle tone and 4 indicating complete immobilisation of the examined limb in flexion or extension [23].

To measure gait function, the Walking Index for Spinal Cord Injury II (WISCI-II) was used to diagnose the improvement in gait function of patients after SCI. The WISCI-II provides an assessment of the level of gait function and is the most sensitive measure of change in gait ability compared to other leading measurement scales. A level of 0 indicates an inability to stand and participate in walking, and a level of 21 indicates an ability to walk without the use of aids, orthoses and assistance from others [53,54].

The Numerical Rating Scale (NRS) was used to determine the level of pain intensity. It is a reliable screening tool that assesses pain severity using an 11-point scale, where 0 means ‘no pain’ and 10 means ‘the worst possible pain one can imagine’ at the time of the test [55].

The Spinal Cord Independence Measure III (SCIM III) was used to assess functional independence after SCI. This scale reliably and accurately assesses the daily functioning skills of patients with spinal cord injury. Version III of the scale consists of 19 tasks grouped into 3 areas, i.e., caring for oneself, respiratory and sphincter control, and mobility. The total score ranges from 0 (total dependence) to 100 (total independence) [56,57,58].

The American Spinal Injury Association Impairment Scale (AIS) was used to diagnose and assess neurological and functional status after SCI. The scale is used to assess the level of motor, reflex, and sensory function and also allows the classification of spinal injury, i.e., A, B, C, D or E; where AIS A indicated complete and AIS B, C and D incomplete SCI [59,60,61].

The Coping Strategies Questionnaire (CSQ) was used to determine coping strategies and provide a subjective assessment of their effectiveness. Seven coping strategies were distinguished: six cognitive and one behavioural. These coping methods are part of 3 factors: (1) cognitive coping, (2) distraction and vicarious coping, and (3) catastrophising and hope-seeking. In addition, the CSQ questionnaire is used to assess the individual’s ability and effectiveness when using strategies to optimise and reduce pain levels [62].

To investigate the location of pain control, the Beliefs about Pain Control Questionnaire (BPCQ) was used. The BPCQ consists of 3 subscales that measure the individual strength of pain control beliefs attributed to: internal factors, the influence of physicians (others) or random external events [62].

The GSES Generalised Self-Efficacy Scale was used to measure patients’ sense of self-efficacy. This scale measures an individual’s belief in their ability to deal effectively with difficult situations [62].

The Hopkins Rehabilitation Engagement Rating Scale—Reablement Version (HRERS-RV) was also used. The questionnaire was translated into Polish. The scale is used to assess patients’ level of engagement in the rehabilitation process and other health care interventions [63].

### 2.4. Procedure

The research procedure was conducted as follows.

Medical physical and neurological examinations were performed. Medical history was obtained and a questionnaire study was performed.

A psychological examination was conducted in the form of a structured interview/questionnaire study. To standardise the research procedure, the psychologist read the questionnaire questions to the study participants and ticked the answers on their behalf. This was due to the motor difficulties of some patients, which led to difficulties marking answers and turning the pages of the sheet.

The aforementioned variables were measured in two stages.

During the patients’ admission to the rehabilitation programme, tests were carried out using the NEO-PI-R Personality Inventory, GSES Self-Efficacy Scale, NRS, CSQ, BPCQ, Ashworth Scale, WISCI II, and SCIM-III.

After completion of the programme, tests were carried out using NRS, Ashworth Scale, WISCI II, SCIM-III, and HRERS-RV.

## 3. Results

### 3.1. Statistical Analyses

The normality of the distribution of the study variables was checked using the Shapiro–Wilk test (Table 3). The results of the test showed distribution close to normal for the impulsivity index in the whole sample and ΔSCIM (differing from comparable scores obtained via the SCIM scale) in the low- and high-neuroticism groups in S0. Due to the numerous outliers, the high values of skewness and kurtosis and the ordinal nature of the Ashworth Scale, analyses were carried out using non-parametric tests. The assumption of homogeneity of variance was met for most cases, except for the improvement scores on the SCIM scale in the S0 group.

The first analyses on pain, spasticity and neuroticism were performed with all participants’ scores. Correlation analysis with Spearman’s rho coefficient was performed. The selection of the non-parametric analysis was based on the ordinal nature of the Ashworth Scale. Therefore, outlier observations (>3SD) within the subscale of neuroticism, e.g., impulsivity, were not removed. The results of the analysis are included in Table 4 and Table 5.

Further calculations were performed for the two rehabilitation groups (S0 vs. S1) and for the low vs. high neuroticism groups. A comparative analysis with the Mann–Whitney U Test was performed. The choice of test was based on the high values of skewness and kurtosis for both variables. The above information is presented in Table 6 and Table 7.

In the first step, the distributions of the variables neuroticism and improvement in rehabilitation outcomes on the WISCI II and SCIM III scales were analysed. The results of the entire study group were taken into account. The results of the Shapiro–Wilk test showed a distribution close to normal for the impulsivity index in the whole sample and ΔSCIM in the low (W = 0.91; *p* = 0.173) and high levels of neuroticism in the S0 group. Due to numerous outlier observations, high values of skewness and kurtosis and the ordinal nature of the Ashworth scale, non-parametric tests were conducted. The assumption of homogeneity of variance was met in almost every case, except for the improvement scores on the SCIM scale in group S0. The results are presented in Table 3.

### 3.2. Neuroticism and Spastic Tension

The results showed that there is an association between neuroticism and spastic tension in the study group (rho = 0.39; *p* < 0.001). This association is positive and moderate. This means that as neuroticism increases, spastic tension increases. See Table 4.

### 3.3. Neuroticism Versus Pain

Spearman’s rho correlation coefficient was also used for the relational analysis between neuroticism and pain. Statistically insignificant results were obtained. The severity of pain is not related to the level of neuroticism of the patients. The data are presented in Table 5.

### 3.4. Gait Function and Functional Independence After Spinal Cord Injury Versus Neuroticism

In order to answer the research question concerning the relationship between rehabilitation outcomes and neuroticism level, a comparative analysis was performed using the Mann–Whitney U test. Statistically insignificant results were obtained. There were no significant differences between the high- and low-neuroticism groups in terms of rehabilitation outcomes. The data are presented in Table 6.

### 3.5. Gait Function (WISCI II) and Functional Independence After Spinal Cord Injury (SCIM III) Versus Type of Rehabilitation

A comparative analysis of rehabilitation outcomes by rehabilitation modality and neuroticism severity showed that the type of rehabilitation (S1 vs. S0) differentiated between patients with low neuroticism in terms of gait function outcomes (*p* = 0.043). The correlation was positive, with moderate strength. This means that people with low neuroticism in the S1 group achieved a higher improvement in gait function. However, the *p*-value after applying the Benjamani–Hochberg correction exceeds the threshold (0.05), so considering the multiple comparisons, the effect obtained should be interpreted with caution. For the other comparisons, the results were not statistically significant. See Table 7.

### 3.6. Self-Efficacy, Beliefs About Pain Control (BPCQ), Coping Strategies for Pain (CSQ) and Neuroticism

In the low-neuroticism group (*n* = 57), distributions resembling a normal distribution were found for two out of three factors for pain control beliefs: internal factors (W = 0.98; *p* = 0.441) and physician influence (W = 0.97; *p* = 0.151), and two out of seven pain management strategies: declaring coping (W = 0.96; *p* = 0.071), increasing behavioural activity (W = 0.97; *p* = 0.145). Similarly, in the high-neuroticism group (*n* = 21), there was a normal distribution for self-efficacy (W = 0.95; *p* = 0.387), pain control belief and re-evaluation (W = 0.94; *p* = 0.202), increasing behavioural activity (W = 0.98; *p* = 0.864), catastrophising (W = 0.94; *p* = 0.180), and praying and hope-seeking (W = 0.95; *p* = 0.361) among those with high neuroticism.

The compared groups were significantly different in size (χ2 (1) = 16.62; *p* < 0.001), and the high-neuroticism group was small in number (*n* = 21), so non-parametric tests were used in further analyses. The results are presented in Table 8.

### 3.7. Self-Efficacy (GSES) and Neuroticism Levels

Median analysis showed that respondents with high levels of neuroticism had lower levels of self-efficacy than those with low neuroticism. The results of the analyses are presented in Table 9.

### 3.8. Pain Control Beliefs vs. Level of Neuroticism

The association between neuroticism level and pain control beliefs in patients after SCI was tested, and the results of each comparison were found to be statistically significant. The strongest, most powerful effect was recorded for internal factors (*p* = 0.001). A moderate effect was recorded for the influence of physicians (*p* = 0.019), when for random external events, the differences are weak (*p* = 0.046). It follows that the level of neuroticism differentiates beliefs about pain control. Patients with low neuroticism were more likely to locate pain control in internal factors than patients with high neuroticism. The opposite association was noted for the influence of physicians and random external events. In this case, patients with high neuroticism were more likely to attribute pain control to physicians’ influences and random events than patients with low neuroticism. The data are presented in Table 10.

### 3.9. Coping Strategies for Pain (CSQ) vs. Level of Neuroticism

Similarly, to verify the answer to the research question regarding the relationship between coping strategies and the level of neuroticism, the results showed that the patients’ level of neuroticism differed between the following strategies: reevaluating (*p* = 0.001), declaring coping praying and seeking hope (strong effects) (*p* = 0.001) and distraction (*p* = 0.039), (moderate effect). The respondents with a low level of neuroticism were more likely to use reevaluating and declaring coping as pain coping strategies than respondents with a high level of neuroticism. In contrast, a stronger preference for the strategies of distraction as well as praying and seeking hope was observed in the group of patients with high levels of neuroticism compared to patients with low neuroticism. The results of the analysis are presented in Table 11.

## 4. Discussion

### 4.1. Neuroticism and Psychosomatic Symptoms After SCI

The present study shows an association of neuroticism with somatic symptoms after SCI, i.e., muscle spasticity. An association of this Big Five personality trait with pain intensity and with the effects of rehabilitation, both the conventional kind and that carried out with RAGT, was not shown.

The association between neuroticism and muscle spasticity has not yet been examined in the literature. It is known from the literature that one of the triggers of spasticity is psychological stress [51,64]. Neuroticism is associated with higher vulnerability to stress and less effective coping in situations involving threat, uncertainty, and anticipation of the occurrence of a harmful factor [31,65,66]. Studies have also shown that increased experimentally induced stress and real harmful psychosocial factors affected the musculoskeletal system through excessive muscle activity seen in EMG and sEMG readings [67,68,69]. Other studies have shown an association of neuroticism with excessive negative stress response only in the area of self-report; physiological symptoms related to the threatening factor were not observed [70].

An element that may explain the association between neuroticism and higher muscle tone is the chronic effect of cortisol (glucocorticoid hormone) [71]. The mechanism involved in the stress response and maintenance of the body’s homeostasis is regulated by the endocrine system, including the hypothalamic–pituitary–adrenal (HPA) axis, which is responsible for, among other things, the regulation and secretion of glucocorticosteroid hormones [72]. When excessive muscle tone is increased under chronic stressors, cortisol is responsible for activating muscle readiness. However, it is important to note that the effects of cortisol on the musculoskeletal system may depend on the amount and duration of exposure to high concentrations of glucocorticosteroid hormones. Long-term exposure of muscles to high levels of these hormones mediating skeletal muscle catabolism and inhibiting the anabolic response leads to muscle atrophy and weakness, and this results in reduced muscle tone [73].

In the studies that have been carried out, no association between neuroticism and pain was confirmed. This contradicts the results of analyses by other researchers, which indicate that neuroticism is associated with increased sensitivity to pain stimuli, a lowered pain threshold, greater tenderness, poorer tolerance, and less effective pain management [38,39,40,41,42,43,44]. The results obtained can be explained by analysing the specifics of study recruitment, the pain coping strategies of study participants, and how neuroticism was defined.

Patients enrolled in the study independently and voluntarily and therefore demonstrated a high level of motivation. In the opinion of the physiotherapists, the patients were assessed as highly motivated, which is confirmed, among other things, by the results of the Hopkins Rehabilitation Engagement Rating Scale—Reablement Version (HRERS-RV) questionnaire (Appendix A). It can be assumed that the invitation to the study was accepted by people who were resourceful, with a sense of empowerment, who were involved in the recovery process and equipped with resources for active coping, including the use of social support. Given the current definition of pain as a personal experience influenced to varying degrees by biological, psychological and social factors, it can be assumed that the abovementioned resources are supportive of effective pain management and/or subjectively perceived lower pain intensity [74].

The results of studies on pain control beliefs showed that patients with high levels of neuroticism presented a so-called strong external type, believing that complaints was outside of their control, and were reliant on the interactions of medical (and paramedical) staff and on random events. Although an external locus of control does not support autonomy in coping with difficulties (potentially leading to increased distress, mood decline, and passivity), it may at the same time be associated with greater trust and systematic adherence to medical recommendations [62,75].

Locus of control may have an important influence on the selection of pain management strategies. Studies have shown that individuals with low (or no) neuroticism traits use more task-focused, active and proactive strategies, promoting effective pain management [62,76]. Individuals with high levels of neuroticism are more likely to use strategies based on praying, hope-seeking, and distraction, adopting avoidance and emotion-focused attitudes. The effectiveness of these strategies in coping with pain is not clear. Some studies indicate that praying positively correlates with pain intensity [76], while others, such as the study by Olivier et al., show that faith and praying minimise pain intensity [77]. It is noteworthy that participants in the studies conducted, in addition to these strategies, used many other strategies based on avoidance and distraction (i.e., redirection as well as inattention). The aforementioned strategies are effective in situations where active action is impossible or associated with increased discomfort [78,79].

The lack of correlation between neuroticism and pain, as opposed to spasticity, can be explained by the fact that pain assessment can be made only on the basis of subjective data that can be operationalised. The severity of spasticity can be determined by objectified methods. It would be worthwhile to improve the methodology for subjective pain estimation by increasing the frequency of estimates, describing the extremes and the middle of the NRS based on the participant’s experience, or offering participants a pain diary to maintain the study period.

This study used the popular Big Five personality model by Costa and McCrae, but there are more recent models that organise the individual components of personality structure slightly differently. In the HEXACO model, emotionality includes only the internalising components of neuroticism, such as anxiety and hypersensitivity, while the externalising aspects, i.e., anger and hostility, belong to the negative pole of agreeableness [80,81]. Another model, the Circumplex of Personality Metatraits (CPM), consists of four matrices (metatraits) with opposite poles that somehow integrate the Big Five factors into specific configurations [81]. The specific configuration of traits can lead to a differentiated picture of functioning among people with similar levels of neuroticism, e.g., difficulties associated with neuroticism can be compensated for by other personality traits, such as extraversion or openness [81].

In addition, factors that could have a significant impact on the results of the analysis of the correlations between neuroticism and pain include the time since the injury occurred, which in this study was 14.40 months in the low-neuroticism group and 15.50 months in the high-neuroticism group; also, low pain intensity in the subjects (M = 3.23, with a scale range of 0–10); other personality dispositions; habituation; and social support. The study did not control for confounding variables such as previous treatment and rehabilitation, or medications taken.

### 4.2. Neuroticism Versus Rehabilitation Modality and Rehabilitation Outcomes

This study found that among people with low neuroticism, patients receiving RAGT (S1) achieved greater improvements in gait function than those receiving conventional rehabilitation with DPT (S0). However, there were no other statistically significant differences in the effects of rehabilitation, either conventional or with RAGT, between SCI patients with high and low neuroticism. These findings are not consistent with the results of other studies, which showed an association between high neuroticism and poorer rehabilitation outcomes [16,41]. The reasons for this difference in results could be the time since injury, during which patients have undergone treatment and rehabilitation; the small size of the sample, the high involvement of all the patients in the rehabilitation process; and potential mutual dependencies between neuroticism and other personality traits.

It is noteworthy that, during the course of the study, each patient was provided with individual and group psychological support using CBT and mindfulness techniques. The psychological support may have contributed to correcting the attitudes of patients with high levels of neuroticism towards rehabilitation interventions, increasing their resistance to stressors and strengthening their sense of self-efficacy in the rehabilitation process.

In addition, positive links between RAGT, proprioception (defined as the sense of position and motion of one’s own body parts and the force generated during movement), gait function, and neurophysiological variables can go beyond the influence of personality variables [82,83].

### 4.3. Limitations and Strengths of the Study

The main limitation of the present analyses was the lack of randomisation. Participants self-referred to the rehabilitation programme, which makes it impossible to consider the study group of people after SCI as representative. The respondents exhibited fairly high motivation to achieve rehabilitation goals (see Appendix A) self-efficacy (see Table 8) and had resources in the form of social support.

In addition, the statistics used only show a correlation between the variables analysed and do not allow the direction of the associations observed to be determined.

In addition, this study would benefit from data on the relationship between the variables analysed in the longer term. In subsequent studies, we plan to re-diagnose the people in this research group to identify any changes in their pain intensity, psychological factors, physical fitness (including independence after SCI), gait function, and spinal cord tension. Through such an analysis, it will be possible to determine the influence of neuroticism on recovery after SCI in the long term. This will have important implications for determining the direction of action.

A strength of this study is that it points out that psychological factors such as personality dispositions may be related to the somatic functioning of patients post SCI. Knowledge of these aspects will facilitate the implementation of effective and targeted treatment interventions including rehabilitation methods, pharmacotherapy and psychological support.

Few studies carried out so far have analysed the relationship between the course and effects of the rehabilitation process of people after SCI and psychological factors. This study fills this gap.

## 5. Conclusions and Clinical Implications

The study showed that neuroticism (as a trait of relatively stable personality dispositions) may be related to the severity of spasticity. However, it was not possible to confirm such an association with pain severity. Neuroticism was also found to be unrelated to rehabilitation outcomes, regardless of its modality (DPT, RAGT).

High severity of neuroticism was shown to differentiate patients in terms of self-efficacy, location of pain control, and use of coping strategies.

It is worth supplementing the rehabilitation and treatment process with an initial psychological diagnosis of patients, enabling the implementation of therapeutic and psycho-educational measures tailored to the patients’ needs and resources. The research conducted justifies psychological interventions aimed at strengthening resilience in SCI patients, e.g., verifying patients’ coping resources, psychoeducation on coping flexibility, goal setting and achievement, and strengthening self-confidence.

## Figures and Tables

**Figure 1 brainsci-14-01153-f001:**
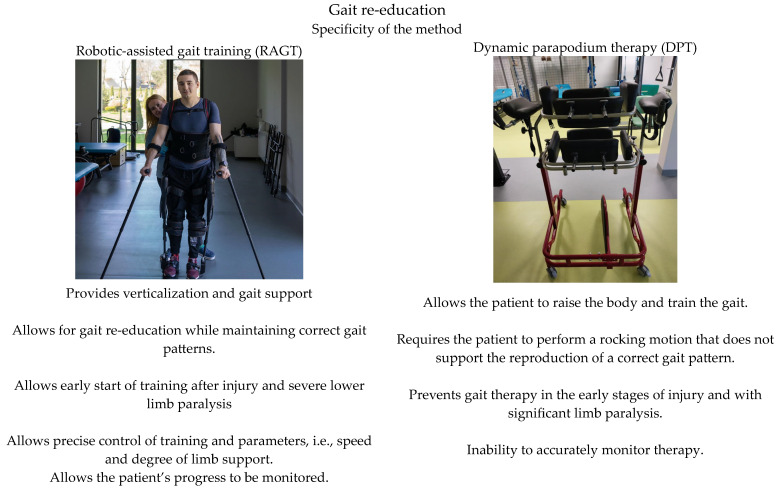
Applications and differences between RAGT and DPT.

**Table 1 brainsci-14-01153-t001:** Distribution of sten scores on the neuroticism scale for low and high levels of neuroticism.

	Low Neuroticism		High Neuroticism	
Sten Scores on the Neuroticism Scale	*n*	%	*n*	%
1	12	21.1%	0	0.0%
2	16	28.1%	0	0.0%
3	19	33.3%	0	0.0%
4	10	17.5%	0	0.0%
7	0	0.0%	5	23.8%
8	0	0.0%	5	23.8%
9	0	0.0%	5	23.8%
10	0	0.0%	6	28.6%

**Table 2 brainsci-14-01153-t002:** Study group characteristics compared to neuroticism level.

Group	Low Neuroticism (*n* = 57; 73.1%)	High Neuroticism (*n* = 21; 26.9%)	*p*
Sex			
Male	43 (75.4%)	18 (85.7%)	0.537 *
Female	14 (24.6%)	3 (14.3%)	
Age			
Median (IQR)	41.00 (25.00)	33.00 (16.00)	0.081
Education			
In the course of education	2 (3.5%)	2 (9.5%)	0.746 *
Elementary	0 (0.0%)	0 (0.0%)	
Vocational	5 (8.8%)	2 (9.5%)	
Secondary	10 (17.5%)	3 (14.3%)	
Higher	40 (70.2%)	14 (66.7%)	
Accommodation			
Countryside	13 (23.2%)	4 (20.0%)	0.312 *
Small town	13 (23.2%)	8 (40.0%)	
Medium-size town	8 (14.3%)	4 (20.0%)	
Big city	22 (39.3%)	4 (20.0%)	
Marital status			
Single	16 (28.1%)	3 (14.3%)	0.420
Informal relationship	11 (19.3%)	19 (4.0%)	
Formal relationship	30 (52.6%)	14 (66.7%)	
Time from accident (months)			
Mean (SD)	14.40 (7.12)	15.50 (8.68)	0.714
Median (IQR)	13.00 (28.00)	13.50 (11.00)	
Cause of injury			
Vehicle accident	14 (25.0%)	11 (52.4%)	0.112 *
Fall < 1 m	3 (5.4%)	1 (4.8%)	
Fall > 1 m	21 (37.5%)	4 (19.0%)	
Dive	3 (5.4%)	1 (4.8%)	
Violence-related trauma	1 (1.8%)	0 (0.0%)	
Body crushing	1 (1.8%)	2 (9.5%)	
Others	13 (23.2%)	2 (9.5%)	
Level of neurological impairment			
Cervical	12 (21.1%)	4 (19.0%)	0.144
Thoracic	25 (43.9%)	14 (66.7%)	
Lumbar	20 (35.1%)	3 (14.3%)	
ASIA			
AIS A	23 (40.4%)	7 (33.3%)	0.572
AIS B/C/D	34 (59.6%)	14 (66.7%)	
Rehabilitation Modality			
S0 (DPT)	13 (22.8%)	11 (52.4%)	0.012
S1 (RAGT)	44 (77.2%)	10 (47.6%)	

Annotation. Differences within age and time since the accident were tested with the Mann–Whitney U Test, while for nominal or ordinal variables, the Chi-Square Test of Independence or (where the assumption of numerosity was broken) the Fisher/Fisher–Freeman–Halton Exact test was used. Fisher’s Exact test was used for a 2 × table, while for larger tables the Fisher–Freeman–Halton Exact test was used. *—Fisher/Fisher–Freeman–Halton Exact test.

**Table 3 brainsci-14-01153-t003:** Basic descriptive statistics together with the result of the normality of the distribution test (Shapiro–Wilk).

Variable	M	Me	SD	Sk.	Kurt.	Min.	Maks.	W	*p*
Entire group ^a^									
Pain	3.23	1.00	2.72	0.87	0.57	1.00	10.00	0.79	<0.001
Neuroticism	80.94	73.00	28.12	0.75	0.03	23.00	154.00	0.94	<0.001
Anxiety	14.27	13.00	6.67	0.55	−0.41	0.00	31.00	0.96	0.003
Hostility	12.44	12.00	5.24	0.57	0.41	0.00	28.00	0.97	0.015
Depression	13.20	12.00	6.07	0.84	0.46	1.00	31.00	0.94	<0.001
Self-Consciousness	15.12	14.00	5.73	0.40	0.04	2.00	28.00	0.97	0.016
Impulsiveness	15.67	16.00	4.82	0.50	0.66	5.00	32.00	0.98	0.107
Vulnerability to Stress	10.80	9.00	6.11	0.56	−0.30	0.00	25.00	0.95	0.001
S0—low neuroticism									
ΔWISCI	0.38	0.00	0.96	2.39	4.79	0.00	3.00	0.47	<0.001
ΔSCIM	4.69	4.00	3.77	0.38	−1.08	0.00	11.00	0.91	0.173
ΔAshworth	0.00	0.00	0.00	-	-	0.00	0.00	-	-
S0—high neuroticism									
ΔWISCI	0.10	0.00	0.32	3.16	10.00	0.00	1.00	0.37	<0.001
ΔSCIM	4.90	2.50	10.68	−0.76	1.60	−18.00	21.00	0.90	0.208
ΔAshworth	−0.05	0.00	0.16	−3.16	10.00	−0.50	0.00	0.37	<0.00
S1—low neuroticism									
ΔWISCI	1.66	0.00	2.72	9.00	2.30	0.00	14.00	0.66	<0.001
ΔSCIM	6.82	6.00	6.58	0.99	0.27	−2.00	24.00	0.90	<0.001
ΔAshworth	0.29	0.00	0.60	0.51	0.59	−1.00	2.00	0.76	<0.001
S1—high neuroticism									
ΔSCIM	6.00	3.50	6.57	1.82	3.77	0.00	22.00	0.81	0.019
ΔWISCI	2.30	0.00	3.30	1.15	0.11	0.00	9.00	0.75	0.004
ΔAshworth	0.20	0.00	0.42	1.78	1.41	0.00	1.00	0.51	<0.001

Annotation. ^a^—the first measurement was included. ΔSCIM—the Delta coefficient of functional independence after SCI; ΔWISCI—the Delta of the WISCI gait function index—II. S1—patients on RAGT, S0—group on DPT; ΔAshworth—the Delta assessment of increased muscle tone (spasticity).

**Table 4 brainsci-14-01153-t004:** Results of Spearman’s rho correlation analyses: neuroticism versus spastic tension (Ashworth scale).

	Spastic Tension	
Variable	Spearman rho	*p*
Neuroticism	0.39	<0.001
Anxiety	0.30	0.004
Aggressive hostility	0.24	0.020
Depressiveness	0.29	0.006
Self-Consciousness	0.28	0.007
Impulsivity	0.32	0.002
Hypersensitivity	0.37	<0.001

**Table 5 brainsci-14-01153-t005:** Results of Spearman’s rho correlation analyses: neuroticism versus pain.

	Pain	
Variable	Spearman rho	*p*
Neuroticism	0.06	0.571
Anxiety	0.07	0.519
Aggressive hostility	0.14	0.171
Depressiveness	0.06	0.565
Self-Consciousness	0.00	0.980
Impulsivity	−0.08	0.453
Hypersensitivity	−0.01	0.903

**Table 6 brainsci-14-01153-t006:** Results of comparative analyses by Mann–Whitney U test in rehabilitation outcomes according to neuroticism level for S0 and S1 groups.

		Low Neuroticism (*n_S_*_0_ = 13; *n_S_*_1_ = 44)			High Neuroticism (*n_S_*_0_ = 10; *n_S_*_1_ = 10)					
Group	Rehabilitation Outcomes	Average Rank	Me	IQR	Average Rank	Me	IQR	U	*p*	η^2^
S0	ΔWISCI	12.35	0.00	0.00	11.55	0.00	0.00	60.50	0.784 ^a^	0.01
	ΔSCIM	11.62	4.00	7.50	12.50	2.50	12.00	60.00	0.784 ^a^	<0.01
S1	ΔWISCI	27.34	0.00	3.00	28.20	0.00	5.25	213.00	0.865	<0.01
	ΔSCIM	27.88	6.00	9.00	25.85	3.50	6.50	203.50	0.712	<0.01

Annotation. ^a^—exact relevance. S1—people rehabilitated with RAGT, S0—group rehabilitated with DPT.

**Table 7 brainsci-14-01153-t007:** Results of comparative analyses by the Mann–Whitney U Test in rehabilitation outcomes by study group for low- and high-neuroticism groups.

		Group S0 (*n_S_*_0_ = 13; *n_S_*_1_ = 10)			Group S1 (*n_S_*_0_ = 44; *n_S_*_1_ = 10)					
Grupa	Outcomes Rehabilitation	Average Rank	Me	IQR	Average Rank	Me	IQR	U	*p*	η^2^
Low neuroticism	ΔWISCI	21.73	0.00	0.00	31.15	0.00	3.00	191.50	0.043 ^b^	0.07
	ΔSCIM	25.62	4.00	7.50	30.00	6.00	9.00	242.00	0.401	0.01
High neuroticism	ΔWISCI	8.80	0.00	0.00	12.20	0.00	5.25	33.00	0.218 ^a^	0.15
	ΔSCIM	10.45	2.50	12.00	10.55	3.50	6.50	49.50	0.971 ^a^	<0.01

Annotation. ^a^—exact significance; ^b^—taking into account Benjamani–Hochberg correction *p* > 0.05, indicating a statistically insignificant effect.

**Table 8 brainsci-14-01153-t008:** Basic descriptive statistics together with the Shapiro–Wilk test score when divided by neuroticism level.

Neuroticism Level	Dependent Variable	M	Me	SD	Sk.	Kurt.	Min.	Max.	W	*p*
Low (*n* = 57)	GSES									
	Self-efficacy	34.93	35.00	3.35	−0.66	0.27	25.00	40.00	0.95	0.031
	BPCQ									
	Internal factors	17.32	17.00	2.84	0.09	0.10	11.00	24.00	0.98	0.441
	Physicians’ influence	15.09	15.00	2.87	−0.47	0.14	8.00	21.00	0.97	0.151
	Random external events	13.32	12.00	3.57	0.30	−0.40	5.00	20.00	0.95	0.021
	CSQ									
	Re-evaluation	13.75	12.50	4.29	−0.06	−1.43	7.00	20.00	0.91	<0.001
	Ignoring	12.93	11.00	4.54	0.73	−0.68	7.00	23.00	0.89	<0.001
	Declaration of coping	19.16	19.00	2.19	0.42	0.02	15.00	25.00	0.96	0.071
	Distraction	17.61	18.00	3.58	0.00	0.69	10.00	28.00	0.94	0.006
	Increasing behavioural activity	21.52	22.00	3.01	−0.02	−0.75	15.00	27.00	0.97	0.145
	Catastrophising	14.11	16.00	4.28	−0.16	−0.95	6.00	24.00	0.93	0.003
	Praying and seeking hope	15.11	16.00	4.80	−0.42	0.17	1.00	25.00	0.95	0.018
	Pain control	2.68	3.00	0.88	0.02	−0.76	1.00	4.00	0.87	<0.001
	Pain reduction	2.13	2.00	0.79	0.24	−0.39	1.00	4.00	0.85	<0.001
High (*n* = 21)	GSES									
	Self-efficacy	30.90	30.00	4.92	−0.07	−0.91	22.00	38.00	0.95	0.387
	BPCQ									
	Internal factors	12.76	12.00	2.76	0.31	−0.73	8.00	18.00	0.95	0.405
	Physicians’ influence	16.71	17.00	2.81	−0.88	0.35	10.00	20.00	0.91	0.052
	Random external events	15.43	16.00	3.74	−0.12	−1.01	10.00	22.00	0.94	0.178
	CSQ									
	Re-evaluation	9.38	10.00	1.63	−0.07	−0.33	6.00	12.00	0.94	0.202
	Ignoring	12.57	11.00	4.07	0.46	−1.24	7.00	20.00	0.90	0.035
	Declaration of coping	16.43	17.00	2.98	−1.18	1.10	9.00	20.00	0.87	0.010
	Distraction	19.19	19.00	2.62	−1.19	2.03	12.00	23.00	0.90	0.029
	Increasing behavioural activity	21.48	21.00	3.72	0.07	1.02	13.00	30.00	0.98	0.864
	Catastrophising	14.48	14.00	4.34	−0.01	−1.07	6.00	21.00	0.94	0.180
	Praying and seeking hope	19.29	20.00	4.08	−0.64	1.30	9.00	27.00	0.95	0.361
	Pain control	2.57	2.00	0.81	0.37	−0.40	1.00	4.00	0.85	0.003
	Pain reduction	2.38	2.00	0.97	0.19	−0.79	1.00	4.00	0.89	0.019

**Table 9 brainsci-14-01153-t009:** The Mann–Whitney U-test—comparison of low and high neurotic patients in terms of self-efficacy.

Variable	Low Neuroticism (*n* = 57)			High Neuroticism (*n* = 21)					
	Average Rank	Me	IQR	Average Rank	Me	IQR	Z	*p*	η^2^
Self-efficacy	44.04	35.00	4.00	25.55	30.00	8.50	−3.24	0.001	0.14

**Table 10 brainsci-14-01153-t010:** The Mann–Whitney U test results for pain control belief indicators according to neuroticism level.

Pain Control Beliefs	Low Neuroticism (*n* = 57)			High Neuroticism (*n* = 21)					
	Average Rank	Me	IQR	Average Rank	Me	IQR	Z	*p*	η^2^
Internal factors	47.34	17.00	4.00	18.21	12.00	4.00	−5.06	<0.001	0.33
Physicians’ influence	35.89	15.00	4.00	49.31	17.00	4.00	−2.34	0.019	0.07
Random external events	36.41	12.00	5.00	47.88	16.00	6.00	−1.99	0.046	0.05

**Table 11 brainsci-14-01153-t011:** Mann—Whitney U-test results for indicators of pain coping strategies according to the level of neuroticism.

Pain Coping Strategy	Low Neuroticism (*n* = 57)			High Neuroticism (*n* = 21)					
	Average Rank	Me	IQR	Average Rank	Me	IQR	Z	*p*	η^2^
Re-evaluation	45.05	12.50	7.75	22.86	10.00	2.00	−3.90	<0.001	0.20
Ignoring	39.69	11.00	7.00	37.17	11.00	7.00	−0.44	0.658	0.00
Declaration of coping	44.59	19.00	3.75	24.10	17.00	2.50	−3.61	<0.001	0.17
Distraction	35.81	18.00	5.00	47.50	19.00	3.00	−2.06	0.039	0.06
Increasing behavioural activity	39.25	22.00	5.00	38.33	21.00	4.00	−0.16	0.872	<0.01
Catastrophising	37.96	16.00	7.00	41.79	14.00	7.50	−0.67	0.501	0.01
Praying and seeking hope	33.69	16.00	7.75	53.17	20.00	4.50	−3.42	0.001	0.15
Pain control	39.79	3.00	1.00	36.88	2.00	1.00	−0.54	0.589	0.00
Pain reduction	37.50	2.00	1.00	43.00	2.00	1.00	−1.03	0.305	0.01

## Data Availability

Data are contained within the article.

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
