# Peer review of "Neuroticism Overestimated? Neuroticism Versus Hypertonia, Pain and Rehabilitation Outcomes in Post-Spinal Cord Injury Patients Rehabilitated Conventionally and with Robotic-Assisted Gait Training"

_brainsci, 2024, doi:10.3390/brainsci14111153_

Round 1
Reviewer 1 Report
Comments and Suggestions for Authors
Strengths
Relevant Topic: The study addresses an important issue in rehabilitation for spinal cord injury (SCI) patients, particularly the role of personality traits like neuroticism in recovery outcomes.
Comprehensive Methodology: The use of various validated instruments (NEO-PI-R, Ashworth Scale, SCIM III, etc.) provides a robust framework for assessing the relationships between neuroticism, spasticity, and rehabilitation outcomes.
Participant Diversity: The inclusion of a significant number of participants (n=110) enhances the reliability of the findings and allows for meaningful statistical analyses.
Areas for Improvement
Clarification of Results: While the results indicate a correlation between neuroticism and spasticity, the lack of significant findings related to pain and rehabilitation outcomes could be elaborated upon. Discussing potential reasons for these outcomes would strengthen the analysis.
Consideration of Confounding Variables: The study could benefit from a deeper exploration of confounding factors that might influence the relationship between neuroticism and rehabilitation outcomes, such as the severity of injury and comorbid psychological conditions.
Longitudinal Aspect: A longitudinal study design could provide insights into how neuroticism impacts recovery over time, rather than just at the point of rehabilitation.
Implications for Practice
Psychological Assessment: The findings suggest the importance of psychological evaluations in rehabilitation settings. Incorporating personality assessments could help tailor rehabilitation programs to individual patient needs.
Holistic Approach to Rehabilitation: The study underscores the necessity of addressing not just physical rehabilitation but also psychological support in patients with high neuroticism to improve overall outcomes.
Conclusion
Overall, this paper contributes valuable insights into the complex interplay between personality traits and rehabilitation outcomes in SCI patients. Further research could expand on these findings by exploring additional psychological factors and their implications for rehabilitation strategies.
Author Response
Dear Reviewer,
Thank you for your valuable suggestions and comments. They were very inspiring for us and have definitely improved our ability to describe the trials analysed. We have tried to respond to all of them and to implement the suggested changes.
comments: Clarification of Results: While the results indicate a correlation between neuroticism and spasticity, the lack of significant findings related to pain and rehabilitation outcomes could be elaborated upon. Discussing potential reasons for these outcomes would strengthen the analysis.
Response: Thank you for this suggestion. We tried to explain the lack of statistically significant association between pain and neuroticism and neuroticism and rehabilitation outcomes.
In the discussion section, we have tried to add information about additional confounding variables.
Comments: Longitudinal Aspect: A longitudinal study design could provide insights into how neuroticism impacts recovery over time, rather than just at the point of rehabilitation.
Response: These are very interesting questions. In the summary we described this question as a guide for future research.

Reviewer 2 Report
Comments and Suggestions for Authors
Dear Authors,
I aprreciated very much this article. The aim of this study was to analyze the association of neuroticism with spasticity and pain in SCI. Additionally, the study investigated the effects of neuroticism on outcomes of conventional rehabilitation (dynamic parapodium) compared to robotic-assisted gait training (RAGT). Here my comments:
ABSTRACT: Please correct "raingin" to "training," and place "RAGT" in parentheses. There are similar small errors throughout the text; please review and correct.
INTRODUCTION: The aim is clearly stated as a question, which I appreciated.
RESULTS: Overall, this article is well-structured, clearly delineating the distinctions between parapodium and RAGT. Including figures and a table highlighting the main differences between these two rehabilitation methods (in goal of rehabilitation project and outcomes) could further enhance clarity.
DISCUSSION: Consider adding that RAGT can improve proprioception, positively influencing spasticity rehabilitation outcomes (doi: 10.3390/jfmk7030053). i.e. RAGT’s potential impact on proprioception, which supports spasticity recovery...
Author Response
Dear Reviewer,
Thank you for your valuable suggestions and comments. They were very inspiring for us and have definitely improved our ability to describe the trials analysed. We have tried to respond to all of them and to implement the suggested changes.
Comment: Please correct "raingin" to "training," and place "RAGT" in parentheses. There are similar small errors throughout the text; please review and correct.
Thank you for bringing these errors to our attention. We have tried to identify and correct them.
Comment: Overall, this article is well-structured, clearly delineating the distinctions between parapodium and RAGT. Including figures and a table highlighting the main differences between these two rehabilitation methods (in goal of rehabilitation project and outcomes) could further enhance clarity.
A table is provided to show the differences between the two gait therapy models (i.e. DPT vs. RAGT).
Comment: Consider adding that RAGT can improve proprioception, positively influencing spasticity rehabilitation outcomes (doi: 10.3390/jfmk7030053). i.e. RAGT’s potential.
Thank you for the very valuable tip and for suggesting an interesting publication. We have included a reference to this publication in the summary.
